# Integrated analysis of miRNA landscape and cellular networking pathways in stage-specific prostate cancer

Shiv Verma[1,2], Mitali Pandey[1¤], Girish C. Shukla[3], Vaibhav Singh[4], Sanjay Gupta[1,2,5,6,7]*

**1** Department of Urology, Case Western Reserve University, School of Medicine, Cleveland, OH, United States of America, **2** The Urology Institute, University Hospitals Cleveland Medical Center, Cleveland, OH, United States of America, **3** Center of Gene Regulation in Health and Disease, Cleveland State University, Cleveland, OH, United States of America, **4** Department of Inflammation and Immunity, Cleveland Clinic, Cleveland, OH, United States of America, **5** Department of Nutrition, Case Western Reserve University, Cleveland, OH, United States of America, **6** Division of General Medical Sciences, Case Comprehensive Cancer Center, Cleveland, OH, United States of America, **7** Department of Urology, Louis Stokes Cleveland Veterans Affairs Medical Center, Cleveland, OH, United States of America

¤ Current address: Vancouver Prostate Center, Vancouver, British Columbia, BC, Canada
* sanjay.gupta@case.edu

**Data Availability Statement:** The miRNA data of all the cell lines were submitted to the GEO public domain portal with accession number GSE119338.

## Abstract

Dysregulation of miRNAs has been demonstrated in several human malignancies including prostate cancer. Due to tissue limitation and variable disease progression, stage-specific miRNAs changes in prostate cancer is unknown. Using chip-based microarray, we investigated global miRNA expression in human prostate cancer LNCaP, PC3, DU145 and 22Rv1 cells representing early-stage, advanced-stage and castration resistant prostate cancer in comparison with normal prostate epithelial cells. A total of 292 miRNAs were differentially expressed with 125 upregulated and 167 downregulated. These miRNAs were involved in pathways including drug resistance drug-efflux, adipogenesis, epithelial-to-mesenchymal transition, bone metamorphosis, and Th1/Th2 signaling. Regulation of miRNAs were interlinked with upstream regulators such as Argonaut 2 (AGO2), Double-Stranded RNA-Specific Endoribonuclease (DICER1), Sjogren syndrome antigen B (SSB), neurofibromatosis 2 (NF2), and peroxisome proliferator activated receptor alpha (PPARA), activated during stage-specific disease progression. Candidate target genes and pathways dysregulated in stage-specific prostate cancer were identified using CS-miRTar database and confirmed in clinical specimens. Integrative network analysis suggested some genes targeted by miRNAs include miR-17, let7g, miR-146, miR-204, miR-205, miR-221, miR-301 and miR-520 having a major effect on their dysregulation in prostate cancer. MiRNA-microarray analysis further identified miR-130a, miR-181, miR-328, miR146 and miR-200 as a panel of novel miRNAs associated with drug resistance drug-efflux and epithelial-to-mesenchymal transition in prostate cancer. Our findings provide evidence on miRNA dysregulation and its association with key functional components in stage-specific prostate cancer.

**Funding:** The research work is supported by United States Public Health Service Grant R01CA108512, VA Merit Award 1I01BX002494 and Department of Defense grant W81XWH-15-1-0558 and W81XWH-18-1-0618 to SG. The funders had no role in study design, data collection and analysis, decision to publish, or preparation of the manuscript.

**Competing interests:** The authors have declared that no competing interests exist.

**Abbreviations:** AGO1, Argonaut 2; ANXA1, Annexin A1; APC, Adenomatous polyposis coli; AR, androgen receptor; BCL2, B-cell lymphoma 2; BCRP, breast cancer resistant protein; BIRC2, Baculoviral IAP Repeat Containing 2; CDKN1B, Cyclin Dependent Kinase Inhibitor 1B; CRPC, castration-resistant prostate cancer; DICER1, Double-Stranded RNA-Specific Endoribonuclease; E2F, E2F family of DNA-binding transcription factors; EGFR, epidermal growth factor receptor; EMT, epithelial-to-mesenchymal transition; ERBB2, erythroblastic oncogene B2; ESR1, Estrogen Receptor 1; FOXO, Forkhead box transcription factor; HEXIM1, HEXIM P-TEFb Complex Subunit 1; HMGA, High Mobility Group Protein; IAP, Inhibitor of Apoptosis; IGF, insulin-like growth factor; IL, interleukins; IPA, Ingenuity Pathway Analysis; JAK/STAT, Janus kinase/signal transducers and activators of transcription; KRAS, Proto-Oncogene; MARK2, Microtubule Affinity Regulating Kinase 2; miRNA, microRNA; mRNA, messenger RNA; mTOR, mammalian target of rapamycin; NF2, neurofibromatosis 2; P-gp/ABCB1, P-glycoprotein; PI3K, Phosphoinositide 3-kinase; PLAG1, Pleomorphic adenoma gene 1; PPARA, peroxisome proliferator activated receptor alpha; PrEc, primary prostate epithelial cells; PSA, prostate-specific antigen; PTEN, Phosphatase and tensin homolog; RNF20, Ring Finger Protein 20; RUNX2, Runt Related Transcription Factor 2; SIRT, Sirtuin; SMARCE1, SWI/SNF Related, Matrix Associated, Actin Dependent Regulator Of Chromatin, Subfamily E, Member 1; SSB, Sjogren syndrome antigen B; SUZ12, Suppressor of Zeste 12; TF, transcription factors; TGFB, Transforming growth factor beta; TSC1, TSC Complex Subunit 1; VEGF, vascular endothelial growth factor; VIM, Vimentin.

## Introduction

Prostate cancer, by far, remains the leading cause of cancer morbidity and mortality in males in the United States, and approximately 20% of these men will develop invasive disease during their lifetime [1]. According to an assessment by the American Cancer Society, in the year 2019, a total of 174,650 men will be diagnosed with prostate cancer and 31,620 prostate cancer-related deaths are predicted [2]. Ethnicity, race, age and family history are some non-modifiable risk factors for prostate cancer, whereas environment and lifestyle factors are modifiable risk factors that contribute to its development [3]. Prostate cancer in humans exhibits a distinctive continuum of features including heterogeneity, multi-focality, inconsistent clinical progression including metastasis to bone, and emergence of androgen-independent disease [4]. The growth of early prostate lesions are androgen dependent, progressing to an androgen-refractory stage, ultimately resulting in patients' death [5]. Current clinical examination including non-invasive imaging, pathological findings, and serum markers are inadequate in prognostication of prostate cancer. Most patients diagnosed with advance-stage disease are treated with androgen deprivation therapy [6], which results in tumor shrinkage [7]. About 70–80% of patients initially respond to this therapy, but the tumor eventually becomes resistant with the emergence of castration-resistant prostate cancer (CRPC) [8]. Although primary prostate cancer is manageable with surgery and/or radiation, however no effective treatment is still available for metastatic CRPC [9]. Recent therapeutic approaches for metastatic forms of prostate cancer are palliative rather than curative. Therefore, understanding the molecular abnormalities during various stages of prostate cancer, in particular CRPC, will lead to the development of approaches for early detection, prevention and therapeutic intervention benefiting patients.

MicroRNAs (miRNAs) are a class of small non-coding regulatory RNAs which are endogenously synthesized molecules about 20–22 nucleotides in length. These miRNAs are well conserved among mammalian species and have essential role in the regulation of ≥60% of human genes [10]. MiRNAs interfere with the targets' expression at the posttranscriptional stage affecting proliferation and differentiation including other cellular pathways. Studies demonstrate that miRNA expression profiles could distinguish between normal and tumor tissues to identify tumors and their subtypes [11–13]. MiRNA can be used as biomarker predicting disease recurrence and clinical outcome in some tumor types [14, 15]. Therefore, miRNAs have emerged as promising tools for diagnosis, prognosis and tumor response in patients. It is expected that an integrated analysis of miRNAs landscape and cellular networking pathways in stage-specific prostate cancer will help in understanding the molecular basis of pathogenesis and is likely to provide information on early diagnosis/prognosis and targets for therapeutic intervention.

In our previous study, we determined the miRNA complement of normal prostate epithelial and stromal cells to obtain a comprehensive assessment of the differential expression of these molecules as potential regulators of various signaling pathways [16]. In the present study, using a microarray platform, we compared the miRNA profiles of various established human prostate cancer cell lines *viz.* LNCaP, PC3, DU145, 22Rv1 with that of normal prostate epithelial cells (PrEc). These cell lines are representative of various stages of prostate cancer including (i) early-stage with androgen sensitivity (LNCaP) (ii) advanced-stage with the loss of androgen receptor (AR) function (PC3 and DU145) and (iii) castration resistant prostate cancer harboring repressed AR (22Rv1). We designed the study to understand the pathological characteristics of prostate cancer linked with miRNA expression at a particular disease stage with emphasis on canonical pathways, upstream regulators, target molecules (miRNA: mRNA) and their regulatory genes followed by clinical validation.

## Material and methods

### Cell culture

Human prostate cancer cell lines LNCaP, PC3, DU145, 22Rv1 and the normal epithelial prostate cells (PrEc) were used in the study. LNCaP cells were established from a supraclavicular lymph node lesion of human prostate adenocarcinoma. These cells possess wild-type AR, are androgen responsive, and secret prostate-specific antigen (PSA) [17]. The PC3 cells were established from the bone metastasis of a grade IV prostatic adenocarcinoma, are androgen-independent and do not express AR or PSA [18]. Similarly, DU145 cells were derived from a brain metastatic site with mutation in AR and are PSA negative [19]. The 22Rv1 cells were derived from a human prostatic carcinoma xenograft after serial propagation in mice with castration-induced regression and relapse of the parental, androgen-dependent CWR22 cells [20]. All cancer cells were maintained in RPMI1640 cell culture medium (Gibco, Gaithersburg, MD) supplemented with 10% fetal bovine serum (Hyclone, GE Healthcare, Pittsburgh, PA), 100 unit's/ml penicillin (Gibco, Grand Island, NY) and 100 µg/ml streptomycin (Gibco, Grand Island, NY). The cells were maintained in an incubator with a humidified atmosphere of 95% air and 5% $CO_2$ at 37˚C. The PrEc cells were cultured in keratinocyte growth medium with supplements including 5 ng ml–1 human recombinant epidermal growth factor and 0.05 mg ml–1 bovine pituitary extract (Life Technologies/Invitrogen, Carlsbad, CA).

### RNA isolation

Total RNA was extracted from LNCaP, PC3, DU145, 22Rv1, and PrEc prostate cell lines using RNeasy Mini kit (Qiagen, Germantown, MD) following manufacturer's protocols. DNA was removed by performing on-column DNase digestion with RNase-free DNase (Qiagen, Germantown, MD). The integrity and size distribution of RNA was examined by agarose gel electrophoresis. From each cell line RNA was isolated in three biological replicates and 3 technical replicates. For quality control, the isolated RNAs were analyzed using the Agilent Bio-analyzer 2100, and RNA with RIN numbers ≥7.2 were used in experiments.

### MiRNA-microarray

MiRNA expression levels were profiled using the Affymetrix array from GenoSensor™ (Tempe, AZ, catlog#1201C) microRNA array version 2007. RNA was isolated from LNCaP, DU145, PC-3, 22Rv1, and PrEc cells and labeled according to the Agilent protocol (version 1.0, April 2007). Arrays were scanned with Agilent Microarray G2565, and Agilent Feature Extraction Software (version 9.5) was used to extract the data. The data summary provides gene names, intensity on three redundancies including average intensity of the triplicates, standard deviation, normalization, and fold changes. Signal intensity measured through fluorescent signal of each chemical detected with either 635 nm or 532 nm and the signals were labeled with either F635 or F532. Bacterial sequences/probes such as gnd and fixB were used as negative and QC controls, which are non-homologous to any targets in the reaction.

### Data analysis and normalization

Normalization represents the normalization ratio of gene signal intensity of a specific gene *versus* global gene signal intensity. The normalization value was then used for the fold change calculation. For quantitative normalization and subsequent data processing GenoExplorer software was used. Quantile normalization of the raw data was performed and differentially expressed miRNAs were identified through volcano plot (p value <0.001) filtering. Hierarchical clustering was accomplished using GraphPad Prism 7 (https://www.graphpad.com) to

distinguish differential miRNA expression profile among the samples. The miRNA data of all the cell lines were submitted to the GEO public domain portal with accession number GSE119338.

## Functional and pathway enrichment analyses

The log2 ratios and P values for each identified RNA in the microarray dataset was inserted into Ingenuity Pathway Analysis (IPA) using the core analysis platform (Ingenuity Systems, Redwood City, CA). The differentially expressed microarray data of prostate cell lines were matched with those in Ingenuity Knowledge Base to generate molecular networks. A list of biological functions overrepresented in the dataset, and determine overrepresented canonical pathways; unmapped RNAs were excluded from further analysis. The p value cutoff of 0.001 was applied. To evaluate the actual over-represented pathways, or to remove the chances of any randomness in data, the more stringent criteria of the Benjamini-Hochberg (B-H) procedure was used.

## Analysis of upstream regulators

The association of miRNA with the upstream regulators, especially transcription factors (TFs), was analyzed using the Ingenuity Pathway Analysis (IPA; Qiagen). The complete dataset with information on Gene ID, FDR, and expression ratio of the original transcriptomic dataset and the datasets corrected by miRNA were used. The annotated microarray was used as background in all analyses. The comparison analysis feature in the IPA Ingenuity® Knowledge Base was used for visualization and analysis which allowed identification of the upstream regulators of differentially expressed genes, prediction of activation Z-score and an overlap P-value for each molecule. The activation Z-score estimates the status of the upstream regulator by using gene expression level of known target gene. A prediction of activation or inhibition (or no prediction) records for the chance that random data generation for significant predictions. Z-scores greater than 2 or smaller than −2 can be considered as significant.

## miRNA and mRNA target interaction

The target of miRNAs was identified using mir-Tar (https://bio.tools/mirtarbase) an integrative web server for identifying miRNA target interactions in humans with Gibbs free energy ΔG (kcal/mol) and alignment score >150.

## Prediction of miRNA target genes

We used CSmiR-Tar database to identify miRNA targets expressed in tissue- and stage- specific disease progression [21]. It also allowed to identify specific target(s) of miRNA using two filter base. The first filter offers target expression in specific tissue, whereas the second filter is disease specific target. Both criteria were used to identify miRNAs and their specific targets in prostate tissue in three stage-specific manner including early stage, advanced stage and CRPC. In particular, these 3 stages comprise of 6 different sub-stages; high-grade prostate intraepithelial neoplasia (putative prostate cancer lesion), prostate adenocarcinoma, androgen-independent prostate cancer and metastatic disease as advance stage prostate cancer, whereas patients exhibiting recurrence after androgen deprivation therapy as CRPC- hormone refractory prostate cancer. Formula used to calculate average normalized score include: Normalized score = Original score-Minimum score / Maximum score-Minimum score; Average normalized score (ANS) = sum of the normalized score from different databases / Number of databases (n = 3).

## Gene ontology analysis

Gene ontology analysis of the retrieved target genes was performed using PANTHER (Protein Analysis through Evolutionary Relationships) (http://www.pantherdb.org) to gain insight in to molecular functional, biological process and cellular component of the target gene products.

## Validation of miRNA data using human patient cohort

The miRNA cell line expression values were matched parallel with human prostate cancer sample cohorts. The following datasets GSE88958, GSE59156 and GSE60117 were used for comparison. The dataset GSE88958 comprise of 21 recurrent and 19 non-recurrent samples obtained from prostate cancer patients. The recurrence was defined as two consecutive serum PSAs greater than 0.2 ng/ml. Patients were followed for until PSA recurrence or at least 4 years for non-recurrent cases. The GSE59156 dataset comprise of 42 samples including 15 benign prostatic hyperplasia, 15 prostate cancer, 9 treatment-resistant prostate cancer and 3 normal prostate cell analogues. The GSE60117 dataset consists of miRNA expression profile of 21 samples from normal prostate and 56 from prostate cancer.

## Statistical analysis

The overlap p-value identified transcriptional regulators having ability to elucidate observed gene expression changes. The overlap p-value measures statistically significant overlap between the dataset genes and the genes that are regulated by an upstream regulator, calculated using Fisher's exact test with p-values <0.01 considered to be significant.

# Results

## Identification of differentially expressed miRNA

The differentially expressed miRNAs in all 4 prostate cancer cell lines including LNCaP, PC3, DU145 and 22Rv1 compared with PrEc are represented in the form of heat map S1 Fig. The red color in the panel shows an increase, whereas the blue color exhibited decrease in miRNA expression with p value <0.001. Results obtained from human prostate cancer LNCaP cells, representative of early-stage cancer, revealed a total of 52 differentially expressed miRNAs, of which, 8 miRNAs were upregulated and 44 miRNAs were downregulated S1 Table. In PC3 cells, 116 miRNAs were differentially expressed including 107 upregulated and 9 downregulated S2 Table. In DU145 cells, 66 miRNAs were differentially expressed with upregulation of 1 miRNA (miR-301; 2.25 fold change) and 65 were downregulated S3 Table. These two cell lines represent advanced-stage cancer. In 22Rv1 cells, a representative of castration-resistant prostate cancer (CRPC), 58 miRNAs were differentially expressed including 9 miRNAs upregulated and 49 downregulated with a fold change cut-off ≥1.5 S4 Table. The data was scrutinized using statistical software graph pad prism (sigma plot) and generated a volcano plot for visualizing differential expression patterns between two different conditions. The statistically significant differentially expressed miRNAs for all 4 cell lines were represented using log2 fold changes (Fig 1).

Next, the comparative analysis of miRNAs was performed between cell line subsets which scooped 1 miRNA (hsa-mir-204; at p value <0.001) linked to LNCaP cells as representative of early-stage prostate cancer. Furthermore, 7 miRNAs (hsa-mir-433-3p, hsa-mir-154, hsa-mir-324 5p, hsa-miR-509-pre, hsa-mir-377, hsa-miR-520f, and hsa-mir-384; p <0.001) were differentially expressed in PC3 cells. In DU145 cells, 12 miRNAs (hsa-mir-127-pre, hsa-miR-488-pre, hsa-mir-451, hsa-mir-138-2-pre, hsa-mir-296, hsa-mir-034b-pre, hsa-mir-194, hsa-mir-023a-pre, hsa-mir-128a-pre, hsa-mir-108, hsa-mir-101, and hsa-mir-026b; p <0.001)

**Fig 1. Volcano plot of miRNA-microarray.** The volcano plot exhibits the relationship between fold-change and significance between the two groups, using a scatter plot view. The y-axis is the negative log2 of P values (a higher value indicates greater significance) and the x-axis is the difference in expression between two experimental groups as measured in log2 space. The blue dots in the figure represents individual miRNAs differentially expressed in four prostate cancer cell lines LNCaP (early-stage), PC3, DU145 (advanced-stage) and 22Rv1 (castration resistant prostate cancer; CRPC). The vertical dotted lines correspond to p value < 0.01 and <0.001, respectively. The horizontal dotted line represents a log 2 fold changes. The left side in the panel corresponds to differentially expressed miRNAs with decreased expression, and the right side in the panel showed miRNA with increased expression.

were differentially expressed. Both cell lines are characteristic of advanced-stage disease progression. Lastly, comparative analysis of 22Rv1 cells, a representative of CRPC, with other prostate cancer cell lines identified 1 differentially expressed miRNA (hsa-mir-520a; p <0.001) (Fig 2).

## Pathway enrichment analysis

MiRNA-microarray data coupled with IPA analysis on LNCaP cells representative of early-stage prostate cancer exhibited three overrepresented pathways. These include (i) drug resistance by drug efflux, (ii) epithelial-mesenchymal transition, and (iii) adipogenesis signaling pathway which were ranked as per their respective–log (p-value) 5.2, 0.62 and 0.33, respectively. Pathway analysis in advanced-stage disease represented by PC3 and DU145 cells identified six overrepresented pathways, and were ranked as per their respective–log (p-value). These include (i) drug resistance by drug efflux (3.41; -log p-value), (ii) adipogenesis pathway (1.29; -log p-value), (iii) epithelial-mesenchymal transition (0.91; -log p-value), (iv) bone metamorphosis (0.83; -log p-value), (v) Th1 pathway (0.67; -log p-value), and (vi) Th2 pathway (0.21; -log p-value), respectively. MiRNA data analysis of 22Rv1 cells characteristic of CRPC identified two overrepresented pathways including (i) drug resistance by drug efflux (9.69; -log p-value), and (ii) epithelial-mesenchymal transition (0.975; -log p-value) pathways (Fig 3A).

Next, the IPA knowledge data in LNCaP cells identified upstream regulators ranked as per their activation Z score. These include (i) argonaute 2 (AGO2; Z score -2.82) (ii) Sjogren syndrome antigen B (SSB; Z score -2.37) (iii) double-stranded RNA-specific endoribonuclease 1 (DICER1; Z score -1.36), and (iv) peroxisome proliferator activated receptor alpha (PPARA; Z score 0.43). The upstream regulators identified in advanced-stage prostate cancer in both PC3 and DU145 cells includes (i) AGO2, (ii) SSB, and (iii) neurofibromatosis 2 (NF2), and (iv) DICER1. The Z activation score in PC3 cells ranged from 2.2 to 3.73, whereas in DU145 cells ranged from -2.16 to -3.77, respectively. The pathway and upstream regulators in 22Rv1 cells, which are representative of CRPC include 5 upstream regulators ranked as per their activation Z score are (i) AGO2 (Z score -2.32), (ii) SSB (Z score -1.99), (iii) DICER1 (Z score -1.11), (iv) NF2 (Z score -1.22), and (v) PPARA (Z score 0.64), respectively (Fig 3B).

## miRNA-upstream transcriptional regulator analysis

The IPA knowledge database identified several miRNAs which post-transcriptionally modulate the expression of upstream regulators during particular disease stage *viz.* early, advance and castration-resistant disease progression (Fig 4A–4P). For instance, the expression of miR-

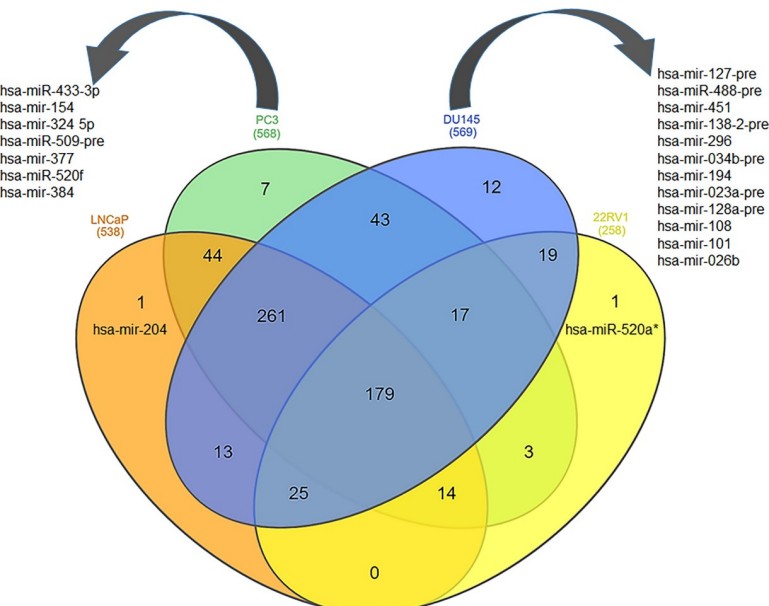

**Fig 2. Prostate cancer disease-specific signature miRNAs.** Venn diagram of the overlapping and unique genes among four prostate cancer cell lines. All four cell lines were represented in orange (LNCaP cells), green (PC3), blue (DU145) and yellow (22Rv1) color and analyzed using p value cut-off < 0.001.

17 was upregulated during early-stage, advance-stage and CRPC (red color) and regulated the expression of AGO2, SSB, DICER1 and PPARA; whereas miR-221 and miR-205 were downregulated (green color) during all stages of disease progression. However, let-7 was differentially

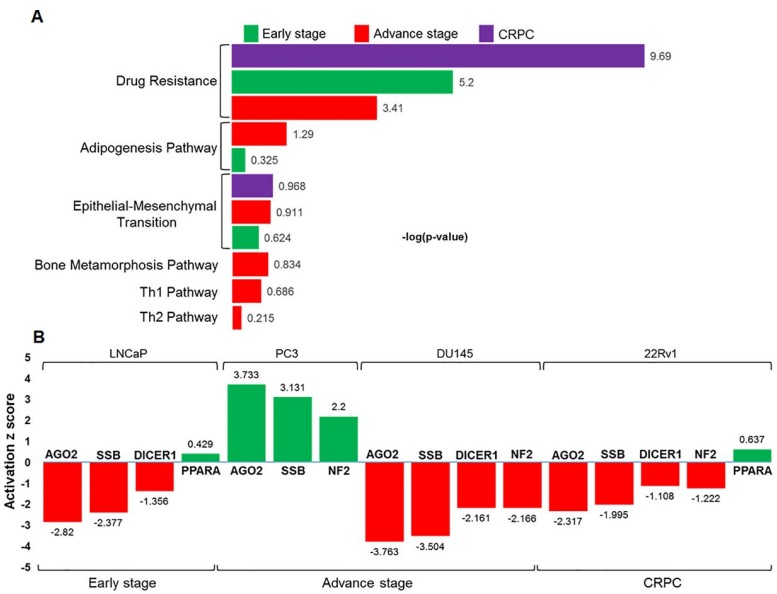

**Fig 3. IPA analysis and overrepresented signaling pathways in various prostate cancer cell lines.** (A) The top scored pathways were scrutinized and ranked as per their–log (p-value). **(B)** Upstream regulators and its activation (green) and inhibition (red) during early-stage, advanced-stage, and CRPC are shown. The values on each bar represent the Z score.

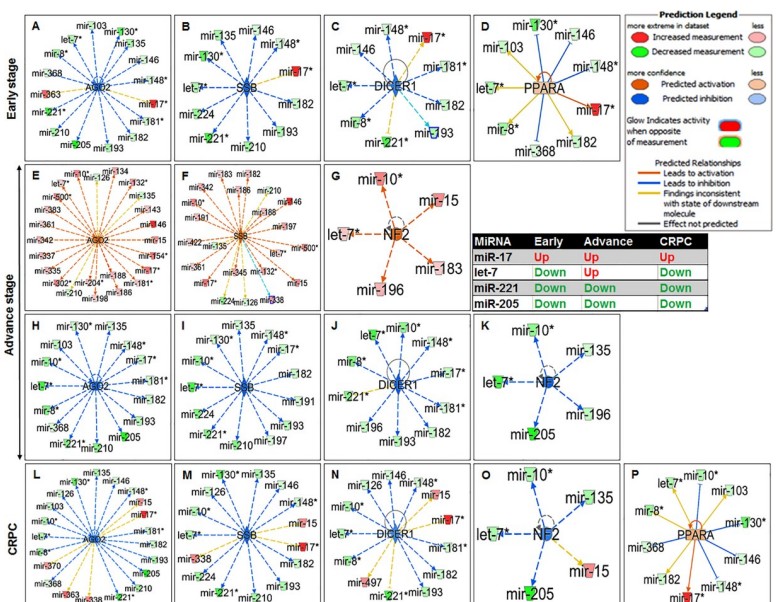

**Fig 4. Inhibition and activation state of upstream translational regulators in relation to miRNAs expression in stage-specific prostate cancer.** The red color showed increased expression of miRNA and green showed decreased (glow indicates activity). The orange and blue color showed predicted activation and inhibition. The dashed and bold arrow indicates and predicts whether it leads to activation (orange arrow) or inhibition (blue line and arrow).

expressed exhibiting downregulation during early-stage and CRPC and upregulation at advance-stage prostate cancer (Fig 4A–4P).

[Fig 5 table image — prostate cancer disease specific miRNA-mRNA target interaction data]

**Fig 5. Prostate cancer disease specific miRNA-mRNA target interaction.** miRNA: mRNA interaction and its target genes categorized on stage-specific manner viz. early-stage, advance-stage and CRPC progression of prostate cancer. Information in the figure include miRNA:mRNA complex with seed alignment score, binding energy (ΔG) of the hybrid complex.

## miRNA and mRNA target interaction of upstream regulators

miRNA Target (miR-Tar) prediction identified the biological functions and regulatory relationships between putative miRNAs and protein coding genes. The miRNA-microarray database coupled with IPA knowledge database deciphered putative miRNAs (miR-10a, miR-17, miR-221, miR-224, miR-130a, and miR-146) and mRNA target interaction of upstream regulators. For example, miR-17 base pairs as G::U wobble at seed sequence with 3'UTR region of AGO2 (seed alignment score of 153 and binding energy ($\Delta$G of -14.1 kcal/mol). Increased expression miR-17 and decreased expression of miR-221 and let-7 influenced the expression of DICER1 activity. The miR-17, miR-221 and let-7 interacted with a transcript of DICER1 and form miRNA:mRNA complex with a seed alignment score of 156, 158 and 141, respectively. The binding energy ($\Delta$G) of the RNA-RNA hybrid complex was -20.40, -16.9, and -14.7 kcal/mol. Furthermore, increased expression of miR-17 and decreased expression of miR-130 also influenced PPARA. The miR-17, and miR-130 interacted with transcript of PPARA and form miRNA:mRNA complex with seed alignment score of 165 and 158, respectively. The binding energy ($\Delta$G) of the hybrid complex was -19.3, and -14.50 kcal/mol. Moreover, lower expression of miR-224 influenced the expression of SSB, the miR-224 binds with 3'UTR region of SSB and form miRNA:mRNA hybrid ($\Delta$G -19.3 kcal/mol) with seed alignment score of 158. Similarly, lower expression of miR-10a influenced the modulation of NF2, it binds with 3'UTR region of NF2 and forms miRNA:mRNA hybrid ($\Delta$G-26.20 kcal/mol) with seed alignment score of 177. The other 2 genes viz. ERBB2 and NF-κB are transcriptionally regulated by miR-205 and form miRNA:mRNA hybrid ($\Delta$G -15.1 and -14.1 kcal/mol) with seed alignment score of 143 and 155, respectively (Fig 5).

## miRNA and gene regulation

The CSmiR-Tar database was applied to identify miRNA targets expressed during stage-specific disease progression. A list of major miRNA regulating stage-specific prostate cancer progression include miR-17, let7g, miR-146, miR-204, miR-205, miR-221, miR-301 and miR-520. Several genes have been identified which were linked to various signaling pathways altered by the above mentioned miRNAs during prostate cancer progression (Fig 6; S2 Fig). Other candidate miRNAs potentially involved in modulating the expression of target genes in prostate cancer are listed in Table 1.

## Gene ontology analysis

Gene ontology analysis of the target genes by PANTHER identified their involvement in different clusters of molecular functions, cellular component, biological process and pathways. Significant results obtained through GO analysis prompt us to uncover stage-specific prostate cancer genes among various target genes. Manual screening of target genes with NCBI Gene database and OMIM prostate cancer genes identified a subset of target genes and pathways which overlap in stage-specific disease progression (Fig 7; Table 1).

## Clinical validation

Next, the parallel matched miRNA profiles of cell lines were validated using miRNA profiles of human prostate specimens. Data revealed significant correlation with clinical specimens such as the upregulated expression of miR-301, miR-520, miR-27a and miR-214 in early-stage and CRPC. Likewise, the expression of miR-214 and miR-18b was consistently upregulated across all prostate cancer cell lines and human patient specimens. In contrast, expression of miR-205 was consistently downregulated in all cell lines and human patient cohort. The expression of

**Fig 6. List of miRNAs and target genes identified using CSmiR-Tar database, miRNAs targets specific genes in prostate tissue and at early, advance and CRPC disease stage.** miRNA regulation of target genes involved in the stage-specific progression of prostate cancer.

miR-021 was in-parallel downregulated in all prostate cancer cell lines but upregulated in human dataset (Table 2).

## Discussion

Dysregulation of miRNAs has been reported in prostate malignancy from early- to advanced-stage and castration resistant disease progression. It is known that miRNAs reside within the fragile regions of the genome, and are implicated to play an important role in cancer progression. Using glass chip-based miRNA-microarray we comprehensively evaluated and profiled miRNA expression using four prostate cancer cell lines as representative of early-stage, advanced-stage and CRPC progression, compared with normal prostate epithelial cells as control (Fig 1; S1 Fig). This study focuses on the comprehensive bioinformatics analysis of miRNA profiles connecting canonical pathways, upstream regulators and their regulatory genes in stage-specific prostate cancer. The associated signaling pathways identified in prostate cancer include adipogenesis, bone metamorphosis, drug resistance drug efflux, epithelial-mesenchymal transition, Th1 and Th2 pathways (Fig 3A), and the upstream regulators *viz*. Argonaute 2 (AGO2), Double-Stranded RNA-Specific Endoribonuclease (DICER1), Sjogren syndrome antigen B (SSB), neurofibromatosis 2 (NF2), and peroxisome proliferator activated receptor alpha (PPARA) (Fig 3B). In addition, miRNA regulatory genes during stage-specific prostate cancer progression suggest the involvement of p53, EGFR-PI3K-Akt, IGF, interleukins, TGFB, VEGF, JAK/STAT, Wnt signaling and their effectors as the most critical genes in prostate cancer *via* upregulation of growth factor receptors, specifically EGFR, or through PTEN inactivation (Fig 7).

Among differentially expressed miRNAs, most interestingly, the expression of miR-301 was upregulated in early-stage and CRPC progression, and this high expression of miR-301 was consistent in both serum and tumor tissue in prostate cancer patients compared to patients with benign prostate hyperplasia [22]. The CSmiR-Tar database identified erythroblastic oncogene B 2 (ERBB2) as a potential target gene of miR-301, across all cancer stages, with a binding site in the 5'UTR of ERBB2 which is structurally related to EGFR (Fig 5) [23]. However, the role of miR-301 in context to its target gene EGFR has not yet been explored. Moreover, EGFR was predominantly upregulated in DU145 and PC3 cells, compared to an androgen-responsive

**Table 1. List of potential miRNAs and its targets interaction expressed in prostate tissue or/and related to a prostate specific disease.**

| miRNAs | High-grade PIN | Early stage cancer | Advance stage | | CRPC | |
| | | | Androgen-independent cancer | Metastatic cancer | Hormone-refractory cancer | Recurrent cancer |
|---|---|---|---|---|---|---|
| hsa-miR-17-3P | TSC1* | HEXIM1 | VIM, AKT3*, AGO2* | SMARCE1, VIM | TCEB1*, PTEN* | PLAG1* |
| has-miR-18b-5P | BCL2 | PTEN, PPARA, BCL2 | PTEN | PTEN | PTEN, SIRT1 | BCL2, PLAG1 |
| hsa-miR-023a-3P | FAS, STS | PTEN | PTEN | PTEN, IRF1, JAK1 | PTEN | PLAG1 |
| hsa-miR-026b | PTGS2, PIM1 | PTEN, PTGS2, KLF6 | PTEN, PTGS2, EZH2 | PTEN, PTGS2, EZH2, WNK1 | PTEN, EZH2, GREB1 | PSAT1, PLAG1 |
| hsa-miR-034b | MYC, MET, CAV1 | HIF1A | XIAP, MAPK1* | CAV1 | XIAP, BCL2 | BCL2 |
| hsa-miR-101 | DUSP1, PTGS2 | PTGS2, KLF6 | EZH2, PTGS2, | EZH2, PTGS2, EGFR | EZH2 | KLF6, SOX9, PLAG1 |
| hsa-miR-127-3P | MAPK8* | CYGB* | PAX2* | PPP2R4* | TP53* | TP53* |
| hsa-miR-128a-3P | EGFR | EGFR | EGFR | SOS1, EGFR, KLF4 | SIRT1, EGFR | EGFR |
| hsa-miR-138-2-3P | RICTOR* | CTNND2* | RASGRP3 | PIK3CD* | STAT3 | PLAG1* |
| has-miR-146-5P | EGFR | ERG | PTGS2, EGFR | PTGS2, EGFR | EGFR | EGFR |
| hsa-miR-154-3P | NKX3-1* | TAC1* | PDZD2* | PPM1A* | SIRT1* | SOX9* |
| hsa-miR-194 | CAV1 | SOCS2, CDKN1B | CDKN1B | CAV1, CDKN1B | PCBD1* | FOXA1* |
| has-miR-204-5P | BCL2 | BCL2 | BIRC2, RUNX2, BCL2 | RUNX2 | SIRT1, PLAG1 | BCL2 |
| hsa-miR-205-5P | ERBB3, BCL2 | VEGFA, PTEN | PTEN | VEGFA, PTEN | AR, PTEN, ERBB3 | AR, ERBB3, E2F1 |
| hsa-miR-221-3P | ANXA1 | ANXA1, CDKN1B | CDKN1B, ESR1, PTEN, AKT3 | CDKN1B, PTEN, RNF20 | ESR1, PTEN, SIRT1 | ERBB3* |
| hsa-miR-224 | PIK3CG | SERPINE1, KRAS/PLAG1 | HSP90AA1, BCL2, CDKN1A* | CDH1, CXCR4, KRAS | BCL2 | BCL2, PLAG1* |
| hsa-miR-296 | CD44* | CD44 | PPP2R4* | CD44 | STAT3* | TP53* |
| hsa-miR-301a-5p | ERBB2 | ERBB2 | ERBB2 | ERBB2 | ERBB2 | ERBB2 |
| hsa-miR-324-5P | TP53 | TP53, SCD | TP53, SMO, SCARB1 | RUNX2* | TP53 | TP53 |
| hsa-miR-377-3P | HMOX1* | EGR1, PTEN | EGR1, PTEN | PITX2, PPM1A | EGR1, PTEN | BMP7* |
| hsa-miR-384 | RICTOR | FGFR1* | MEF2C* | WNK1* | KLK3* | KLK3* |
| hsa-miR-451a | CAV1* | TSC1* | RASGRP3* | CAV1* | NPEPPS* | NPEPPS* |
| hsa-miR-488-3P | NKX3-1* | NKX3-1* | FSCN1* | IGF1* | FSCN1* | NKX3-1* |
| hsa-miR-433-3P | TFAP2A, MAPK8 | KRAS, MAPK4 | KRAS,TFAP2A | | AR*, BCL2* | AR* |
| hsa-miR-509-3P | APOD* | ZFX | PTEN* | BCAR1 | HLA-A | KLF6* |
| hsa-miR-520a | MARK2 | ESR1 | ESR1 | SUZ12 | ESR1 | ESR1 |
| hsa-let7b-3P | SENP1* | PLAG1* | NR3C1*, PLAG1* | WNK1 | C1orf52 | SOX9*, PLAG1* |
| hsa-let-7g-5P | DUSP1, MYC | EDN1 | EDN1, CDKN1A | EDN1 | CASP3, AHR, HMGB1 | TP53* |

*Average normalized score of from 3 database; DIANA-microT, https://www.miranda.org/, miRDB, and Targetscan which was ranked with high to low p value.

Formula used to calculate the above

Genes without star are validated experimentally

LNCaP cell line [24, 25], in which both EGFR and ERBB2 were expressed [26]. EGF/EGFR signaling activates ERK and Akt pathways [27], that markedly promotes cell migration and survival. Taken together, the above data support that dysregulation of ERBB2/EGFR activates downstream pro-oncogenic pathways which include AKT-PI3K-mTOR pathways, critical for cancer cell migration and proliferation. Based on the above information, miR-301 plays a critical role in prostate cancer progression and may either be developed as a prognostic marker or therapeutic target [28].

Another miRNA, miR-146b/146 exhibited higher expression (5.38/4.7 fold) during advanced-stage prostate cancer. Consistent with the analysis performed in 42 patient samples including 15 benign prostatic hyperplasia, 15 prostate cancer, and 9 treatment-resistant prostate cancer, the expression of miR-146b was upregulated during CRPC progression (GSE59156). Similar overexpression pattern was found in human patient cohort [29], and in androgen-refractory prostate cancer DU145 and PC3 cells [29]. With reference to its target interaction, miR-146 binds at 3′UTR coding region of EGFR during early-stage, PTGS2 in advance-stage and EGFR in CRPC progression (Fig 5). Previous reports suggested that miR-146-5p represses EGFR expression through binding to its 3'-untranslated region [30], and that the reconstitution of miR-146b-5p may be useful for the treatment of invasive prostate cancer [31].

The expression of miR-205 was downregulated in all prostate cancer cell lines progressively in stage-specific disease progression (S1–S4 Tables). Similar pattern of downregulation of miR-205 was noted in prostate cancer tissues from Caucasian-American and African-American men, compared to adjacent normal tissue [32]. It is speculated that downregulation of miR-205 is inversely correlated with methylation status of miR-205 promoter and locus (S2 Fig). miR-205 actively targets the 3'UTR of some key genes such as AR, BCL2, ERBB3, PTEN, and VEGF-A that play important roles in cancer cell invasion and metastasis (Fig 6). During

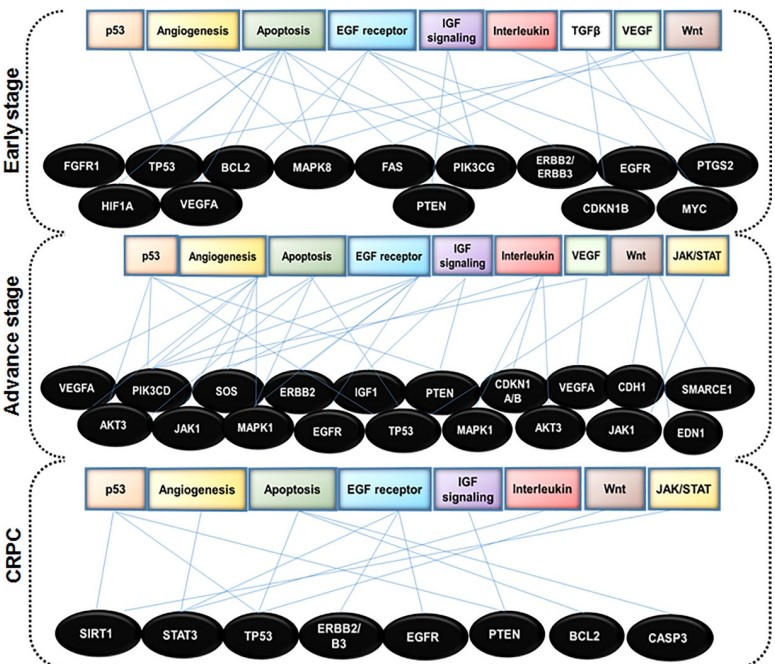

**Fig 7. Gene ontology analysis of target genes.** Target genes were found to be involved in the molecular function of various signaling pathways in stage-specific prostate cancer progression.

**Table 2. List of potential miRNAs and their expression (Up or Down) were matched parallel with human prostate cancer sample cohorts.** The expression of miRNAs from human patient's samples were derived from three GEO databases including the recurrent and non-recurrent cases (GSE88958), with primary prostate cancer stem cells (GSE59156), and the miRNA expression profile of prostate cancer compared with normal prostate tissue (GSE60117).

| miRNAs | LNCaP | PC3 | DU145 | 22Rv1 | Clinical data | Benign | Cancer | CRPC/recurrence | Log Fc | Adjusted p value | P value |
|---|---|---|---|---|---|---|---|---|---|---|---|
| **hsa-miR-301** | Up | Up | Up | Up | GSE59156 | Up | Up | Up | | 0.914 | 0.005 |
| | | | | | GSE88958 | NA | NA | Down | -0.790 | 0.409 | 0.0139114 |
| | | | | | GSE60117 | NA | Down | NA | -0.331 | 0.00466 | 0.001 |
| **hsa-miR-205** | Down | Down | Down | Down | GSE59156 | NA | NA | NA | **NA** | NS | NS |
| | | | | | GSE88958 | NA | NA | Down | -0.905 | 0.502 | 0.0301838 |
| | | | | | GSE60117 | NA | Down | NA | -1.045 | 2.62E-02 | 6.24E-03 |
| **hsa-miR-146** | Down | Up | Down | Down | GSE59156 | Down | Down | Up | | 0.914 | 0.023 |
| | | | | | GSE88958 | | | Down | -0.4737 | 0.388 | 0.0116907 |
| | | | | | GSE60117 | | Up | | 1.06419 | | |
| **hsa-miR-222** | Down | Up | Down | Down | GSE59156 | Down | Up | Up | | 0.914 | 0.055 |
| | | | | | GSE88958 | | | Up | 1.08054 | | |
| | | | | | GSE60117 | | Up | | 0.04842 | | |
| **hsa-miR-520a** | Up | Up | Down | Down | GSE59156 | Down | Up | Up | | 0.914 | 0.001 |
| | | | | | GSE88958 | | | Up | 1.2789 | | |
| | | | | | GSE60117 | | Down | | -0.0833 | | |
| **hsa-miR-18b** | Up | Up | Up | Up | GSE59156 | Up | Up | Up | | 0.914 | 0.003 |
| | | | | | GSE88958 | | | | 0.65415 | | |
| | | | | | GSE60117 | | Up | | 0.09036 | | |
| **hsa-miR-130a** | Down | Up | Down | Down | GSE59156 | Down | Up | Up | | 0.914 | 0.002 |
| | | | | | GSE88958 | | | Up | 0.88362 | | |
| | | | | | GSE60117 | | Down | | -0.3678 | | |
| **hsa-miR-224** | Down | Down | Down | Down | GSE59156 | Down | Up | Up | | 0.914 | 0.003 |
| | | | | | GSE88958 | | | Up | 0.63174 | | |
| | | | | | GSE60117 | | Down | | -0.07 | | |
| **hsa-miR-021** | Down | Down | Down | Down | GSE59156 | Up | Up | Up | | 0.914 | 0.003 |
| | | | | | GSE88958 | | | Up | 1.07261 | | |
| | | | | | GSE60117 | | Up | | 1.07261 | | |
| **hsa-let-7f** | Up | Up | Down | Down | GSE59156 | Up | Up | Down | | 0.505 | 0.000 |
| | | | | | GSE88958 | | | Up | 0.65392 | | |
| | | | | | GSE60117 | | Up | | 0.77012 | | |
| **hsa-let-7b** | Up | Up | Down | Down | GSE59156 | NA | NA | NA | | 0.815 | 0.001 |
| | | | | | GSE88958 | | | Up | 0.8535 | | |
| | | | | | GSE60117 | | Down | | -0.526 | | |
| **hsa-let-7e** | Down | Up | Down | Down | GSE59156 | Up | Up | Down | | 0.914 | 0.049 |
| | | | | | GSE88958 | | | Down | -0.6196 | | |
| | | | | | GSE60117 | | Down | | -0.2765 | | |
| **hsa-let-7g** | Down | Up | Down | Down | GSE59156 | Up | Up | Down | | 0.914 | 0.019 |
| | | | | | GSE88958 | | | Up | 0.49065 | | |
| | | | | | GSE60117 | | Up | | 0.59126 | | |
| **hsa-let-7d** | Down | Up | Down | Down | GSE59156 | Up | Up | Up | | 0.914 | 0.010 |
| | | | | | GSE88958 | | | Down | -0.643 | | |
| | | | | | GSE60117 | | Up | | 0.36822 | | |
| **hsa-miR-27a** | Up | Up | Up | Up | GSE59156 | Up | Up | Up | | 0.914 | 0.019 |
| | | | | | GSE88958 | | | Up | 0.61732 | | |
| | | | | | GSE60117 | | Down | | -0.0831 | | |
| **hsa-miR-214** | Up | Up | Up | Up | GSE59156 | Up | Up | Up | | 0.914 | 0.007 |
| | | | | | GSE88958 | | | Up | 0.66996 | | |
| | | | | | GSE60117 | | Down | | -0.7541 | | |

*(Continued)*

**Table 2.** (Continued)

| miRNAs | LNCaP | PC3 | DU145 | 22Rv1 | Clinical data | Benign | Cancer | CRPC/recurrence | Log Fc | Adjusted p value | P value |
|---|---|---|---|---|---|---|---|---|---|---|---|
| has-miR181a | Down | Up | Down | Down | GSE59156 | Down | Up | Down | | 0.914 | 0.006 |
| | | | | | GSE88958 | | | Down | -0.5123 | | |
| | | | | | GSE60117 | | Up | | 0.497 | | |

advanced-stage and CRPC, miR-205 targets the same set of genes (PTEN and VEGF-A) along with AR, and E2F1 transcription factor. Moreover, downregulation of miR-205 was inversely correlated with BCL-2 upregulation [33], an anti-apoptotic oncoprotein required for prostate cancer progression [34]. Mechanistically, miR-205 targets the ERBB3 receptor and inhibits Akt activation; the expression of ERBB3 was correlated with biochemical recurrence. Conclusively, miR-205 is a novel molecule having potential to serve as a predictive biomarker and therapeutic target in prostate cancer [35].

Another miRNA, miR-221 was downregulated in the miRNA-microarray dataset (S1–S4 Tables). Downregulation of miR-221 was documented in LNCaP cells as early-stage cancer [36, 37], and underlines the involvement of miR-221 in the expansion and maintenance of CRPC phenotype [38]. The miR-221-3p targets genes at early stage such as Annexin A1 (ANXA1), Cyclin Dependent Kinase Inhibitor 1B (CDKN1B), and multiple genes during advance-stage cancer such as CDKN1B, PTEN, Ring Finger Protein 20 (RNF20), Estrogen Receptor 1 (ESR1) and AKT3 (Fig 6; Table 1). In CRPC, miR-221 targets Adenomatous polyposis coli (APC), and AR (Fig 6). Moreover, in context to unique/signature disease stage specific miRNAs, upregulation of miR-204 was associated with early stage whereas downregulation of miR-520a was identified as signature for CRPC (Fig 6). The miR-204 acts as tumor suppressor in human cancer by targeting BCL2 at an early-stage, and, thus protecting cancer cells from undergoing apoptosis as a resistance to androgen deprivation therapy [39]. During advanced-stage, miR-204 targets Runt Related Transcription Factor 2 (RUNX2), Baculoviral IAP Repeat Containing 2 (BIRC2). RUNX2 is a bone-specific transcriptional regulator, aberrantly expressed in metastatic prostate cancer cells [40] and BIRC2, also known as Inhibitor of Apoptosis (IAP) plays key role in drug resistance and survival of cancer cells through inhibition of apoptosis [41]. During CRPC progression, miR-204 targets SIRT1 gene, which is involved in cancer drug resistance [42] by FOXO1 deacetylation [43, 44].

Another miRNA, miR-520a binds to the 3' UTR of target genes including Microtubule Affinity Regulating Kinase 2 (MARK2), Estrogen Receptor 1 (ESR1) at early-stage cancer and Suppressor of Zeste 12 (SUZ12) during advance-stage and ESR1 during CRPC progression of prostate cancer (Table 1). The role of MARK2 with respect to prostate cancer was not yet been studied, while previous reports showed that ESR1 play protective roles in prostate cancer development [45], though still unknown. Moreover, the role of SUZ12 was pivotal, indeed mutation in SUZ12 suppressed tumor growth in prostate [46].

The miR-17 is an oncomiR overexpressed in all stages of prostate cancer. The miR-17~92 cluster targets 3'UTR of tumor suppressor PTEN mRNA [47]. Studies suggest that miR-17~92 cluster promoter binds to E2F1-3 and activated its transcription in several cell types through a negative feedback loop [48]. The miR-17~92 expression is frequently upregulated in prostate cancer cells, resulting in E2F1-3 depletion thereby evading apoptosis. Besides miR-17 target genes altered during early-stage cancer include TSC1 and HEXIM1. In advance-stage miR-17 targets SMARCE1, VIM, AKT3, AGO2 and PLAG1; whereas TCEB1 and PTEN are major targets of miR-17 during CRPC. Studies suggest that miR-17 overexpression leads to downregulation of STAT3 expression, which in turn, suppresses proliferation in human prostate cancer

LNCaP cells. STAT3 is a member of STAT protein family and a transcription factor phosphorylated by receptor-associated Janus kinases (JAK), playing a critical role in many biological processes including cellular proliferation.

Let-7g miRNAs has been recognized as a tumor suppressor that plays an important role in various cellular processes and loss of its expression is associated with cancer development and progression [49]. Our previous publication has shown that the Let-7 family members are significantly expressed in both prostate epithelial and stromal cells [16]. Members of the let-7g miRNA targets and negatively regulates KRAS and MYC oncogenes, chromatin associated non-histone proteins *viz*. HMGA1, HMGA2, cell cycle regulator CDK6, and Lin28B that promotes cell proliferation, invasion and metastasis [50]. In fact, replacement of let-7g member's expression could be developed as a valid therapeutic option in certain types of tumor. Conclusively, the expression of above set of miRNAs including miR17, let-7g, miR-146, miR-204, miR-205, miR-301, miR-221 and miR520 remarkably modulate the expression of several target genes which include AR, ANXA1, BCL2, CDKN1B, EGFR, ESR1, MARK2, RUNX2, PTEN, SIRT1, SUZ12, VEGF-A, TSC1, HEXIM1and others in disease specific stage influencing various signaling pathways in proliferation, invasion, angiogenesis, migration and evasion of apoptosis.

miRNA-microarray expression data coupled with the IPA knowledge database identified associated signaling pathways and its interacting miRNAs such as miR-181a, miR130a, and miR-328 as predominant molecules associated with drug resistance drug efflux pathway in prostate cancer. The expression of 130a/b, miR-181a, (downregulated in early-stage, upregulated in advance-stage, and downregulated in CRPC) and miR-328 (upregulated in advance-stage). Moreover, the drug resistance-drug efflux pathway showed that miR-130a/b, and miR-181a were associated with member's ABC transporter family including P-glycoprotein (P-gp/ABCB1), and breast cancer resistant protein-(BCRP) (S2 Fig). Overexpression of ABC transporter proteins is associated with the resistance of a variety of chemotherapeutic drugs [51]. Likewise, miR-146a was differentially upregulated in PC3 cells and was associated with Th1 and Th2 signaling pathway (S2 Fig). Considerable evidence showed that miR-146a was a potent inhibitor of Th1 differentiation and cell proliferation of human T cells, and dysregulation of miR-146a contributed to the pathogenesis [52].

Another microRNA, miR-200b which was significantly downregulated in CRPC (S4 Table), was associated with epithelial-to-mesenchymal transition (EMT) pathway, and directly associated with Jagged1-Notch1 signaling (S2 Fig). Increasing evidences indicate that Notch1 expression was influenced by miR-200, and together they play an essential role in EMT progression. In consequence, EMT promotes cancer metastases and closely correlates with increased stemness and drug resistance. Moreover, the role of miR-200 warrant further investigation particularly during CRPC disease stage. Conclusively, the role of miRNA which includes miR-130a, miR-181, miR146 and miR-200 are considered to be important in context to prostate cancer.

Transcription factors (TFs) and miRNAs are essential regulators that directly or indirectly controls each other expression and effector genes through feedback and feed-forward loops [53]. TF and miRNA co-regulation is predominant in biological systems and any perturbation to this co-regulation alters the expression of upstream regulators. Earlier, inactivation/activation of upstream translational regulators has not received much attention in prostate cancer. In the present study the expression of AGO2, SSB, and NF2 is downregulated in early-stage cancer and upregulated in advance-stage cancer; whereas PPARA was overexpressed and DICER1 was inhibited across all stages of prostate cancer (Fig 3B). Documented evidence showed that higher expression of AGO2 was directly associated with prostate cancer cell proliferation, apoptosis, and cell cycle regulation [46]. Expression of miR-146 was upregulated only during advance-stage cancer, while miR-17 was upregulated, miR-221, miR-205, let-7g was

downregulated across all disease stage (S1–S4 Tables). Consequently, AGO2 was inactivated during early-stage and CRPC and activated at advance-stage (Fig 3B). Indeed, the interaction between miRNA: mRNA showed that miRNAs (miR-17, miR-205, miR-224, miR-let-7g, miR-10a and miR-130a) binds to the 3'UTR region of upstream regulators (AGO2, PPARA, NF2, DICER1 and SSB), which suppresses their posttranscriptional regulation either at the translation step of degrading the transcript by deadenylation. We successfully found target genes and pathways which have some degree of overlap between stage-specific disease progression (Figs 6 and 7; Table 1). However, the above facts and data warrant further functional research and validation either in preclinical model or established cell lines, which may lead to novel findings in context to prostate cancer, and biomarkers either prognosis or diagnosis.

In conclusion, miRNAs have emerged as essential modulators of immunity, cellular physiology and cancer. Our findings across prostate cancer cell lines identified potential candidate miRNAs involved in regulating target genes essential for the development and progression of prostate cancer. Most miRNAs have multiple targets at different stage of prostate cancer, thus it is not surprising that different targets may be involved in cancer cell apoptosis, cell differentiation and proliferation. Along with miRNAs, we have identified some potential upstream regulators which display critical role in prostate carcinogenesis, and further investigation on them may open a new avenue in prostate cancer research. We have identified a panel of consistently dysregulated miRNA in the present study and thus, validation of these promising candidate miRNAs in experimental studies and in a prospective cohort will not only define their exact role in prostate carcinogenesis but may also develop them as potential prognostic and diagnostic markers. We are currently analyzing the functional role of some miRNAs in prostate cancer. It is our understanding that further studies on these differentially expressed miRNAs will lead to a better understanding of mechanisms mediating the development and progression of prostate cancer.

## Supporting information

**S1 Fig. Heat map of miRNA-microarray.** Expression of miRNAs differentially expressed and assessed in microarray analysis of RNA isolated from four different cell lines of prostate cancer (LNCaP, PC3, DU145, 22Rv1) compared with control cell line of prostate cancer (PrEc). The red color depicts high and green color showed a lower level of expression at p value <0.01. (TIF)

**S2 Fig. Drug resistance by drug efflux.** The internetworking relationship with miRNAs and plasma membrane protein P-glycoprotein (Pgp-plasma membrane protein). The miRNA-130a and miR-181a are downregulated in this pathway (green color) and linked with Pg and BCRP (breast cancer resistant protein). miR-133a and miR379 are involved in regulating the expression of MRP2. While miR-298, miR27a, miR331-5p and miR-130a are involved in regulating the expression of poly-glycoprotein (P-gp). **Epithelial mesenchymal transition pathway.** The miR-200b was downregulated during early-stage prostate cancer and was involved in inhibiting Jagged-2 (JAG2), one of the NOTCH ligands. **Adipogenesis pathway.** The expression of miR-326 was upregulated during the metastatic stage of prostate cancer, (red color) and involved in regulating the expression of CCAT/enhancer binding protein β (C/EBPβ), directly linked with the nuclear hormone receptor peroxisome proliferator-activated receptor-gamma (PPAR-γ). B**one metamorphosis signaling pathway.** The expression of miR-140, miR-145, and miR-155 was upregulated, along with miR-140 and miR-145 were associated with modulation of gene SOX9, and miR-155 was involved in regulating the gene FOXO3A. **Th1 pathway.** In this pathway, the expression of miR-146a showed a higher level of expression (red color) and modulated the expression of NF-κB signaling. **Th1 and Th2 pathway.** In this pathway,

the expression of miR-146a showed a lower level of expression (green color) and may modulate the expression of NF-κB signaling.
(PDF)

**S1 Table. List of differentially expressed miRNAs derived from LNCaP cells lines statistically significant as P < 0.001.**
(XLSX)

**S2 Table. List of differentially expressed miRNAs derived from PC3 cells lines statistically significant as P < 0.001.**
(XLSX)

**S3 Table. List of differentially expressed miRNAs derived from DU145 cells lines statistically significant as P < 0.001.**
(XLSX)

**S4 Table. List of differentially expressed miRNAs derived from 22Rv1 cells lines statistically significant as P < 0.001.**
(XLSX)

## Acknowledgments

We are thankful to Dr. Nagalakshmi Nadiminty, PhD, University of Toledo for careful review and suggestion on the manuscript.

## Author Contributions

**Conceptualization:** Shiv Verma, Sanjay Gupta.

**Formal analysis:** Girish C. Shukla, Sanjay Gupta.

**Funding acquisition:** Sanjay Gupta.

**Methodology:** Shiv Verma, Mitali Pandey.

**Software:** Vaibhav Singh.

**Supervision:** Girish C. Shukla, Sanjay Gupta.

**Validation:** Shiv Verma, Girish C. Shukla.

**Visualization:** Shiv Verma, Sanjay Gupta.

**Writing – original draft:** Shiv Verma, Sanjay Gupta.

**Writing – review & editing:** Mitali Pandey, Girish C. Shukla, Sanjay Gupta.

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
