## [Decision Letter · Decision Letter 0]

7 Oct 2019

Integrated analysis of miRNA landscape and cellular networking pathways in stage-specific prostate cancer

PONE-D-19-22454

Dear Dr. Gupta,

We are pleased to inform you that your manuscript has been judged scientifically suitable for publication and will be formally accepted for publication once it complies with all outstanding technical requirements.

In your amendments to the manuscript,please provide a detailed description and specific justification of the human sample datasets used in this study in the Methods section.

With kind regards,

Klaus Roemer

Academic Editor

PLOS ONE

Additional Editor Comments (optional):

Reviewers' comments:

Reviewer's Responses to Questions

**Comments to the Author**

1. Is the manuscript technically sound, and do the data support the conclusions?

Reviewer #1: Partly

Reviewer #2: Yes

Reviewer #3: Yes

2. Has the statistical analysis been performed appropriately and rigorously? 

Reviewer #1: I Don't Know

Reviewer #2: Yes

Reviewer #3: Yes

3. Have the authors made all data underlying the findings in their manuscript fully available?

Reviewer #1: Yes

Reviewer #2: Yes

Reviewer #3: Yes

4. Is the manuscript presented in an intelligible fashion and written in standard English?

Reviewer #1: Yes

Reviewer #2: Yes

Reviewer #3: Yes

5. Review Comments to the Author

Reviewer #1: Authors in this manuscript performed detailed miRNA profiling and their targeted pathways of normal prostate epithelial cells and prostate cancer cell lines derived from advanced prostate cancer with androgen-sensitive or insensitive. The study is innovative and of significance. However,all these prostate cancer cell lines were derived from patients with metastatic or castration resistant diseases. There is no early stage prostate cancer cell line.

Reviewer #2: The manuscript by Verma et al describes an integrated analysis of miRNA expression in various prostate cancer cell lines representative for different stages of this pathology and attempted validation in existing human prostate cancer sample datasets. The concept is valid and the manuscript is well written. There are a few minor issues that would need to be addressed before publication:

As the authors recognize in the introduction, prostate cancer is characterized by a high degree of heterogeneity and multi-focality. Therefore, a more detailed description of the human sample datasests is warranted, and justification of their choice for analyzing these specific datasets given that multiple datasests are publicly available. It would have been much more informative if the authors could perform their own collection of well characterized human samples and analyze those in the same manner as they did for the cell lines.

Reviewer #3: This article covers an up to date topic regarding miRNA landscape and prostatic carcinogenesis, making a step further to future prognostic and diagnostic markers by not only linking miRNA dysregulations to prostatic adenocarcinoma, but focusing on cellular networking pathways in finding novel miRNAs abnormalities associated with different stages of prostatic cancer.

The conceptualization of the study offers a clear methodology and continues a previous published work of the authors, now targeting representative cell lines for prostatic adenocarcinoma at early disease stage (LNCaP cells), advanced stage (PC3 cells and DU145 cells) and castration resistance stage (22Rv1 cells). The studied miRNAs profiles are known for their relation with human prostate cancer lines, but stage-specific miRNAs changes in prostate cancer are not known. The present study brings new comprehensive bioinformatics analysis of miRNA profiles connecting canonical pathways, upstream regulators and their regulatory genes for different stages of prostatic adenocarcinoma.

The manuscript is well written with concise ideas using qualitative references, presenting a well thought design with no language mistakes noted. The usage of statistical software graph pad prism, GenoExplorer software, Ingenuity systems, mir-Tar web server, CSmiR-Tar database, PANTHER gene ontology analysis and validating cell culture results with miRNA profiles from human patient cohort, make the results reliable and well represented with figures and tables containing easy to read data.

By finding target genes and pathways that overlap between stage disease progression at a certain degree, this study could lead to understanding the development of prostatic cancer, with further research being necessary for correlating miRNAs and upstream regulators with cancer cell apoptosis, cell differentiation and proliferation, therefore, better leading to better diagnosis and treatment management.

6. PLOS authors have the option to publish the peer review history of their article (what does this mean?). If published, this will include your full peer review and any attached files.

Reviewer #1: No

Reviewer #2: No

Reviewer #3: No

---

## [Editor Report · Acceptance letter]

17 Oct 2019

PONE-D-19-22454 

Integrated analysis of miRNA landscape and cellular networking pathways in stage-specific prostate cancer 

Dear Dr. Gupta:

I am pleased to inform you that your manuscript has been deemed suitable for publication in PLOS ONE. Congratulations! Your manuscript is now with our production department. 

With kind regards,

on behalf of

Dr. Klaus Roemer 

Academic Editor

PLOS ONE